# Nonlocality in Quantum Mechanics Portrayed as a Human Twins’ Metaphor

**DOI:** 10.3390/e25020192

**Published:** 2023-01-18

**Authors:** Salomon S. Mizrahi

**Affiliations:** Physics Department, Federal University of São Carlos, Via Washington Luiz km 235, São Carlos 13565, SP, Brazil; salomonsmizrahi@gmail.com or salomon@df.ufscar.br

**Keywords:** non-locality, action-at-a-distance, twins’-metaphor, paradoxes

## Abstract

Avoiding the use of mathematical formalism, this essay exposes the quantum mechanics phenomenon of nonlocality in terms of a metaphor involving human twins, focused on their hands’ dexterity attribute.

## 1. Introduction

It is widely admitted that the first axiomatic quantum theory is owed to Niels Bohr – he belongs to the select team that confronted the scientific challenges of the time –, when he proposed a novel path, diverging from the previous theories, to explain why matter is essentially stable. In 1913 he elaborated a model for the atom, postulating: (1) the stability of the electron orbiting the hydrogen nucleus and, (2) that the emitted radiation (the spectral lines) is due to electron jumps between quantized orbits. Going back further in time and analyzing the progress of scientific discoveries, one spots the interval 1900–1913 – that can be recognized as the *early quantum age* –, beginning when Max Planck communicated his, perhaps, most important contribution to science, the correct formula for the black-body radiation, ingrained with the energy quantization as multiples of a new standard number, *h*, which, later, was perceived as a fundamental universal constant, thence referred, in his honor, as Planck’s constant or Planck’s *quantum of action* (Its value in SI units is $h=6.62607004\times 10^{-34}$ m${}^{2}$kg/s.). Nevertheless, the numbers resulting from Bohr’s atomic model – and also from subsequent similar but modified ones –, were only partially satisfactory because the data collected from measurements were not reproduced accurately by the prevailing theory, and this was due, somehow, to the formal and conceptual deficiencies of the models. From then on, until 1925, the use of recent instrumentation and the adoption of new methods for analyzing the data contributed to a remarkable progress in experimental procedures regarding precision and refinement. Gazing at the theoretical stage, despite the several new proposals, the data fittings were still unsatisfactory, eventually due to conceptual limitations. The big leap in the theoretical framework happened between 1925 and 1928. Diverging from the old paradigm, an original route of thinking blossomed, with the adoption of modern revolutionary concepts, sustained by novel mathematical tools, bringing forth a new theory which displayed sound and quite more accurate reproducibility of the current data; it then became known as *Quantum Mechanics* (The designation “Quantum Mechanics” was first adopted by Max Born in a paper published in 1924 [1], which preceded the historical paper by Werner Heisenberg [2], although it was still not a Matrix Mechanics. Nevertheless, one year later, the Quantum Matrix Mechanics was solidly established within the academic milieu after the publication of the articles [3,4,5,6,7]. See the book [8] that contains these and other fundamental papers; those originally published in German are translated into English.) or *Wave Mechanics* (in 1924, L. de Broglie launched the idea of wave-particle duality and, in 1925, E. Schrödinger invented, or discovered, the seminal equation that bears his name and introduced the – then enigmatic – wavefunction); consult [9] for the complete collection of Erwin Schrödinger’s articles on wave mechanics. A history of the intellectual trajectory of the statistical interpretation of quantum mechanics was given by Abraham Pais [10], and for a complete narrative about Schrödinger’s scientific achievements see [11,12,13].

The disclosed discoveries entailed the quick acceptance of the revolutionary paradigm by the most influential academic circles. The modern quantum theory endorsed its success through the description, with high degree of rigor, of atomic structures and to also unravel and outline correctly new properties of matter. Nevertheless, despite its accomplishments, the logic inherent to the theory did not pass unshaken (in opposition to the classical areas of phenomenology such as mechanics, thermodynamics, and electromagnetism, whose casual logical inconsistencies were rarely questioned) because within the paradigm, embroiled into the formalism, a curious and very peculiar phenomenon could not fit into the current logic, and as such, it became a challenge to the prevalent common sense. Later, in 1935, ten years after the first publications that established quantum mechanics, such issue was directly addressed by A. Einstein, B. Podolski, and N. Rosen (EPR) [14], that was promptly contested by N. Bohr [15], while supported by E. Schrödinger [16]. That specific phenomenon can be referred as the *non-locality* riddle, which is currently assumed to be a feature inherent to the nature of matter. It exhibits its more pronounced peculiar properties in the atomic realm. From that time until nowadays, it has been recognized as a cornerstone of quantum theory because it turned out to be an essential feature for understanding the stability of atoms and structured matter, from elementary particles to large molecules. Its influence in science and philosophy is profound in that it stimulated several different interpretations of quantum mechanics, sparking heated debates among its creators and, even today, it is at the root of a variety of controversial epistemic discussions [17,18,19,20].

To ascertain the correct properties of atoms, for being confirmed by a sound theoretical formulation, it is essential to get a good data fitting, and for this end, quantum mechanics needed to equip itself with a mathematical set of definitions and theorems proper to the three-dimensional *Euclidean space* ($E_{3}$), because the position vector $\vec{r}$ is an argument in Schrödinger’s wavefunction. This manifold is already familiar to us, primarily because it is where we live and secondly, for being the locus where natural phenomena - macro, micro, organic and inorganic - evolve. Notwithstanding, from the study of the structural properties of atoms and electromagnetic radiation, the physicists understood that $E_{3}$ could not be enough to explain the finer details arising from more recent data. The reality compelled them to append at $E_{3}$, a new stealthy degree of freedom, the intrinsic and discrete *spin* [21,22]; this inherent property ubiquitously present in fundamental particles, until then, went unobserved. In what follows, the formalism based on classical mechanics attached to wave physics had to be reformulated to interweave the spin with $E_{3}$, whose aim was to get a more embracing theory, able to explain the more recent measured properties, and propose experiments to unravel fresh ones. This assignment was achieved after introducing a mathematical structure known as the discrete Hilbert space (H). The enhanced theory became the *quantum mechanics with spin* or *Pauli’s formulation* (worked out by Wolfgang Pauli [23]) and the expanded space $E_{3}\times H$ stretched the horizon for describing more accurately atomic and subatomic phenomena. To represent formally matter and light, in this expanded “universe”, old mathematical objects had to be adapted, and original ones, as state vectors, wavefunctions, operators, projectors, matrices, etc., were introduced, becoming indispensable mathematical tools in the novel theory. Back to the issue of non-locality, since its advent, the various astounding interpretations defied the dominant common sense of that time and, in reaction, a commitment imbued the physicists with the target to embrace, with perseverance, the task of solving the puzzle. They aspired to expose the cause of the controversy to heal the malaise it was causing within the scientific community. Next, by dispensing the mathematical formalisms, I introduce a metaphor emulating the problem.

## 2. The Problem

The non-local action at a distance (AAD) is a quantum mechanical riddle that shows up whenever one part of a bipartite system affects and shapes its distant complementary, without direct interaction and traveling signal. It is worth reminding that in classical and in ‘regular’ quantum physics, a signal is a disturbance propagating in space and time, like a sound wave, an electromagnetic pulse, a gravitational wave, or a particle carrying a sort of information, as a photon, or else. Although quite different from classical physics, notwithstanding, it concerns the issue of cause and effect. From a logical standpoint, within the quantum framework, non-locality is an intriguing way to convey a pseudo signal, compared to deliveries achieved, for instance, by an electromagnetic pulse to be picked up by an antenna. This subject annoyed Einstein and his collaborators, prompting them to write a scientific article where they discussed the phenomenon [14], in which they disclosed an instance that, due to its proper construction, led to a paradox. Their verdict was: quantum mechanics could not be accepted as a complete theory. In the concluding paragraph of the EPR paper, they state, *Although we have shown that the wave function* [describing some physical system] *does not provide a complete description of physical reality, we leave open the question of whether such a description exists. However, we believe that such a theory is possible*. Thus, according to their logical way of thinking, Einstein and colleagues admitted that, at a forthcoming time, the quantum theory would be replaced (or complemented) by a more comprehensive one.

It is paramount to emphasize that Einstein never refuted quantum mechanics *per se*, he considered it as a useful theory, being an essential tool for describing accurately the matter properties. Nevertheless, due to his epistemological and philosophical conception of nature, Einstein’s beliefs clashed with the interpretation advocated by the founding fathers of quantum mechanics, thus leading to a hermeneutics paradox. Till the end of his life, Einstein did not accept quantum mechanics as a definitive theory, and according to his own words,

“Probably never before has a theory been evolved which has given a key to the interpretation and calculation of such a heterogeneous group of phenomena of experience as has the quantum theory. In spite of this, however, I believe that the theory is apt to beguile us into error in our search for a uniform basis for physics, because, in my belief, *it is an incomplete representation of real things* [my italics], although it is the only one which can be built out of the fundamental concepts of force and material points (quantum corrections to classical mechanics). The incompleteness of the representation is the outcome of the statistical nature (incompleteness) of the laws. … [Max] Born statistical interpretation of the quantum theory is the only possible one. The ψ function does not, in any way, describe a condition which could be that of a single system; it relates rather to many systems, to “an ensemble of systems” in the sense of statistical mechanics. If, except for certain special cases, the ψ function furnishes only *statistical* data concerning measurable magnitudes… the ψ function does not, in any sense, describe the condition of one single system” [24].

In front of the displayed scenario, I hope that, even without following a rigorous approach, a metaphor may contribute to exposing more comprehensibly the paradox. Other discussions on the subject are found, for instance, in [17,25,26,27,28], while, as declared by F. Wilczek (he is a Nobel prize awarded, 2004) in the last paragraph of his article, *What is Quantum theory* [29], in commemoration of the 75th year of the quantum mechanics birth, “To summarize, I feel that after seventy-five years – and innumerable successful applications – we are still two big steps away from understanding quantum theory properly.”

## 3. The Twins’ Metaphor

The events occur within a classical universe, i.e., where the only conceived physical space is Euclidean $E_{3}$. In one of its habitats, the Earth, a geneticist resolves to induce a specific mutation in a human zygote, and he then implants the embryo in a woman’s womb. Later, at the end of the gestation period, she delivers same-sex twins, having an inherent peculiar attribute related to their hands’ dexterity. One of the twins will be manifestly left-handed while the other will, necessarily, be right-handed, with no chance of being ambidextrous. It is admitted that the genetic mutation is not faulty. After the twins’ birth, the geneticist has no viable approach to identify their hands’ dexterity, so he must wait until they reach a suitable age to carry out an assured verification. At their birth, the twins are set apart, one remains on Earth and the other is sent far away, let us say, to planet Mars, which is, as on Earth, a place where the events obey the laws of classical physics. As the geneticist’s goal is to unravel the dexterous hand, when the boys become two years old one (for instance, the Earthling) receives a pencil and a sheet of paper to scribble. The use of the right hand indicates that he is a right-hander, then for sure, without the need to test his twin brother on Mars, the geneticist deduces, unmistakably, that he is left-handed. The finding does not depend on the distance between the twins, and it is admitted they never maintained previous communications. According to their comprehension of nature, Einstein, Podolski and Rosen affirmed that there is an element of physical reality in the child’s (Martian) hand dexterity, i.e., from the result of the applied test on the Earthling twin, the Martian hand dexterity can be determined without testing, a process usually referred to *local realism*. The genetic manipulation imposed a deterministic attribute on the twins, the uncertainty before the test was due to the geneticist’s cognitive condition, in no way related to the twins’ intrinsic state.

In another scenario, one can imagine a distinct universe $U_{q}$ where the events are not only subjected to the classical laws of physics and Laplace’s determinism (the $E_{3}$ space), but where the objects, from micro up to macroscopic sizes, are under the influence of a natural extra degree of freedom – ruling even over living organisms independently of their size – accountable for the appearance of very peculiar characteristics as, *ambiguity* and *indeterminacy*. Hence, repeating the same procedures as in $E_{3}$ world, in the $U_{q}$ the geneticist manipulates the zygote such to endow the twins with a specific *indeterminacy* related to their hands’ dexterity, inducing a dichotomous feature innate to the Hilbert space $H_{2}$. In this stage, one child is kept on Earth while his twin is transported to Mars.

What distinguishes the twins born in the $E_{3}$ world from those born in the $U_{q}=E_{3}\times H_{2}$? In the latter, before the very moment a test is performed, each child cannot be considered being decisively right-handed or left-handed, whereas in $E_{3}$ the hand dexterity is definitive from the day the mutated embryos were implanted. In $U_{q}$ the twins remain in the very peculiar state of *undecidability* that lasts until the very moment a test is realized on any one. Regarding the dexterity quality only, it can be said that they live in an intertwined *limbo* state.

On the day the Earthling child is to be tested (there is no more connection between twins and no more communication between geneticists), one could concede that (although it is incorrect in the real world of atoms, as to be explained in the subsection) instead of allowing the child to choose the hand to grip a pen and scribble, the decision about the dexterous hands rests on the geneticist’s will. After choosing the right hand, the child will immediately become a right-hander, alternatively, choosing the left hand will make the child left-hander. Decisively, this procedure endows the geneticist with the capability to determine, according to his own will and on the spot, the twins’ hand dexterity. As a reaction to the Earthling geneticist’s choice, the twin on Mars becomes *instantaneously* left-handed or right-handed, the opposite hand dexterity of his Earthly brother, without interference or knowledge of the geneticist on Mars. Thus, after testing one of the twins, the geneticist rescues both children from their limbo state, attributing to each one a definitive hand dexterity. Therein consists of the essence of the instantaneous and deterministic non-local AAD, it concerns the geneticist’s capability to influence the child on Mars without any physical contact or classical communication between the twins and between the geneticists. Ought to their conceptual and logical understanding of Nature, Einstein, Podolsky, and Rosen called this phenomenon *spooky action at a distance*, as they deemed it to be logically inadmissible.

The question that remained unanswered under the classical physics rationale is: how could it be that without any kind of conventional communication channel – the absence of interaction between the twins since their separation – a local action affects, right away and without time delay, a physical feature at a large distance? This non-local effect became the enigmatic question not only for Einstein, Podolski, and Rosen but also for other contemporary physicists such as de Broglie and Schrödinger himself. Yet, on the other side, in the defense of the thesis that quantum mechanics is indeed a complete theory, notwithstanding its predictions being probabilistic, were aligned high-intellectual status physicists such as Niels Bohr, Max Born, Wolfgang Pauli, and Werner Heisenberg, founders of the so-called *Copenhagen interpretation* of quantum mechanics. The main advocate of their cause was Bohr, he criticized the EPR assertions and conclusions [14] through another paper, [15], under the same title. According to the reasoning of the Copenhagen group, the concept of reality in the EPR paper does not apply to quantum mechanics, for its members the physical reality is accomplished at the very moment the children get off the limbo state to acquire hand dexterity. More precisely, *a dynamical variable does not acquire a real value until it is measured* and factually, in quantum theory only the data obtained through measurements reflect reality. In this connection a paper by C. A. Fuchs and A. Peres has the suggestive title *Quantum Theory Needs No ’Interpretation’* [27].

As already mentioned, there is a subtle and precise inaccuracy in the above narration, to be exposed here. In the world of atoms and photons, a fictitious geneticist does not have the prerogative of choosing the hand to be touched for turning a twin into a right-hander or left-hander. Factually, the process is not deterministic, hands dexterity occurs at random, and the geneticist is only a partially active agent. At the very moment a geneticist touches, let us say, the head of one child, he, *de facto*, triggers the exit of both twins from the limbo state, and hand dexterity is confirmed haphazardly and definitively. The process occurs with the same chance as when a fair coin is tossed, and the outcome is expected to be head or tail at a 50%:50% rate. Nevertheless, the spooky AAD is still present, but quite differently from a coin, for which the possible outcomes, head or tail, cannot be separated and sent afar.

The word spooky expresses the sudden dissolution of the undecidability at the very moment the Earthling, or the Martian, geneticist touches the head of the child under his care, independently of the distance separating the twins. The attribution of hand dexterity, at random, to one child, affects the hand dexterity of his twin without the reception of a signal (as an electromagnetic pulse or a photon) and without the action of a field force, as the instantaneous AAD in Newtonian gravitation. According to physicist M. Gleiser, “He [Einstein] spent the rest of his life trying to exorcise the quantum demon, without success … Even weirder, this ability to tell one from the other persists for arbitrarily large distances and appears to be instantaneous. In other words, quantum spookiness defies both space and time.” [30].

### Randomness and Instantaneity of AAD Hinders Oddities

Presupposing that the Earthling (or Martian) geneticist chooses, according to his free will, the dexterous hand of the twin under his control, this implies that the suppression of the undecidability does not occur at random (*as it does in fact*), thence, the geneticists become active agents, and as such, they can take advantage of their skills to make practical use of twins, as long as they remain in limbo state. On the other hand, remarking that the distance between Earth and Mars varies over time, the transit time of an electromagnetic signal traveling at c=300.000 km/s goes between 4.3 and 21 min thence, to eliminate that time delay in exchanging messages, they decide to make use of the twins’ state of undecidability. Here it is not hard to spot an oddity in such a procedure.

Let us envision a dystopic world where geneticists are capable of producing thousands of twins in a limbo state regarding hand dexterity; they then manage to, after their birth, separate them, one half remains on Earth, and their twin brothers are dispatched to Mars, where, there too, dwells a team of geneticists. Those living on Mars are also active agents, and likewise their Earthling colleagues they can decide about a child to be right-handed or left-handed, a choice that will also affect his twin brother on Earth. In this way, the geneticists decide to create an instantaneous AAD communication channel according to a preset binary code. They use the children to send and receive messages through a sequence of bits: right-handed = 0 and left-handed = 1, which is binary alphabet. Each Earthling and Martian geneticist separates the children into two groups, the $A_{E}$ group (Earthlings) and the $A_{M}$ group (Martians); each child of the $A_{E}$ group is in a limbo state with his twin brother in the $A_{M}$ group. Within the $A_{E}$ group the geneticist numbers the twins from $1_{E}$ to $N_{E}$ and their brothers on Mars are numbered from $1_{M}$ to $N_{M}$; the same procedure goes for the children in groups $B_{E}$ and $B_{M}$. In what follows, whenever an Earthling geneticist decides to send a message to his partner at Mars he will touch the hands of, for instance, 15 children, those labeled $1_{E}$ to $15_{E}$, from $A_{E}$ group; as so, the message is transmitted to Mars instantaneously and the geneticist will be aware of it after checking the hands dexterity of the twins $1_{M}$ to $15_{M}$. For example, if the Earthling geneticist sends the binary codified message 011010111001000, the Martian geneticist tests the children $1_{M}$ to $15_{M}$, that are no more in a limbo state, and he annotates who is right-handed and who is left-handed, he will then get the sequence 100101000110111, which is exactly the complementary sequence of bits sent by the Earthling geneticist; then, by just changing 1⇄0 the original transmitted codified sequence is retrieved. Thereafter, by its turn, pursuing the same procedure, the Martian geneticist uses the $B_{M}$ children group to transmit a message to the Earthling geneticist, who tests the children in the $B_{E}$ group to read the received message. According to this protocol, adapted for this AAD non-local communication channel, it takes almost “zero time” to activate each distant bit (a superluminal speed is not forbidden, meaning that the AAD may, virtually, transmit an out-of-limbo order at a “faster than the light” pace). Nevertheless one should consider a time lapse between two AADs; in brief, the time it takes to send a full message from Earth to Mars will depend only on the sum of the time lapses.

Even so, that instantaneous communication channel is not feasible because, as already mentioned, the geneticists do not have the prerogative to choose the hand dexterity of the twins, and whenever one of them precedes the other in touching a twin’s head, this action takes both children, immediately, out-of-limbo state, nonetheless their definitive hand dexterity is probabilistic (for instance, 50%:50%), implying that a sequence of conveyed AAD cannot form a meaningful message. For a more technical explanation, the reader may go to section 12-14-1 in [31], in which, by expounding experiments with photons, the authors demonstrate that despite the non-local AAD, the causality principle is not violated; according to the principle of causality, in two related sequential events, the former is the cause that affects the latter.

Conforming to quantum mechanic’s formalism, the out-of-limbo AAD (or state reduction after a measurement) cannot be a sort of signal (look that each twin can be used either as a sender or as a receiver) traveling at a finite speed because it will eventually lead to conflicting AADs, thus falling into a paradox. How does this oddity arise? Assuming that an AAD is a signal traveling at a finite speed, and at the very moment the Earthling geneticist touches the head of the child under his charge, the twins should go out-of-limbo state, but at different times, one immediately and the other later, when the signal reaches him. After testing the Earthling twin, the geneticist notices that he became a right-hander, and due to the earlier genetic manipulation, the signal attaining the Martian twin should make him a left-hander. Still, while the signal is on the fly toward Mars the local geneticist, unaware of what his Earthling counterpart did, touches the head of ‘his’ twin, which promptly becomes, by chance too, right-handed and, concomitantly, an AAD signal travels towards his Earthling brother to turn him into a left-hander, but he already became a right-hander! The question is, what could happen to the Martian twin at the moment the signal conveyed from Earth reaches him to make him a left-hander when he already became right-handed? *Mutatis mutandis*, one can also ask, what is going to occur, for the same reason, with the Martian twin? These questions reveal the existence of a non-soluble conflict of AADs. Nevertheless, this contradiction was never observed in the real world of atoms and particles and, in theory, it could not happen because the mathematical structure of quantum mechanics is consistent and self-contained, thus this sort of paradox is definitively avoided as an out-of-limbo AAD occurs instantaneously, i.e., there is no propagating signal. Already in 1936, the question of state measurement of one of two entangled subsystems with the AAD determining the state of the other was approached by Schrödinger [32].

In summary, differently from signal-tradings between inertial frames of Special Relativity (SR) [33], it is admitted that for a two-part entangled state of quantum mechanics, no effective signal is emitted or shared whenever a measurement executed on one part affects immediately its faraway counterpart. Even if one dares to admit that the non-local AAD phenomenon may be a signal that exceeds the speed of light in a vacuum, it has no connection with SR, as signals (electromagnetic pulses or photons) propagate without exceeding that speed, yet carrying an amount of energy that can be totally or partially absorbed by atoms or molecules, following thence that the non-local AAD has no place in SR. Results from experiments done in that direction can be found, for instance, in [31,34,35,36,37,38,39] and references therein.

## 4. Epilogue

The data emerging from numerous experiments involving micro-objects such as electrons, photons, atoms, etc. collected within labs [34,35,36,37] as well through communication between satellite (sender) and lab (receiver) [38,39], are correctly explained by the $E_{3}\times H$ formalism, which is at the base of the non-relativistic quantum mechanics, thus supporting the hypothesis about a “ghostly randomness” in nature. This peculiarity is related to the fact that subsystems, located afar from each other, which interacted in the past, during a small lapse of time, still keep a very peculiar correlation – a limbo state – that may be portrayed as a fragile and intangible virtual thread intertwining them. This kind of correlation can be disrupted either by a measurement on one of the subsystems or through the action of some uncontrollable external interference commonly attributed to the environment, and known as *decoherence* [40,41,42].

The cited experiments were motivated by two previous theoretical advancements: (1) In the 1950s, D. Bohm came up with a thought experiment based on the use of a discrete dichotomic variable to emulate the non-local AAD effect [43]; (2) a few years later, J. S. Bell [44] disclosed a set of mathematical inequalities involving classical measurable variables, enclosing *hidden parameters*, to test the possibility to emulate quantum predictions. Both contributions stimulated experimentalists to undertake successful designs (the usual procedure to “reveal-the-reality” consists of planning an experimental setup and gathering data from many repeated *runs*, such to build up “good statistics” to be compared with theory [36], or, otherwise, confront with a numerical simulation [45]) that confirmed Bohm and Bell’s contributions.

Last but not least, it is opportune to remark that randomness and probabilities permeate all phenomena described by quantum mechanics, among which are: (1) the tunnel effect (see, for instance, [46] for an extensive review) and (2) the appearance of a diffraction pattern when a beam of electrons goes through a crystal [47]; later, (3) the interference effect was confirmed for electrons – emulating Young´s double-slit experiment – and reported in [48]. This ingenious experiment consisted of a few thousand electrons falling one after the other on a screen until the buildup interference pattern turns visually revealed. As so, the particle-wave duality of the matter was uncovered, not only through diffraction but also by interference. 

## Data Availability

Not applicable.

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
