# Peer review of "Nonlocality in Quantum Mechanics Portrayed as a Human Twins’ Metaphor"

_entropy, 2023, doi:10.3390/e25020192_

Round 1

Reviewer 1 Report

In the manuscript entitled "Non-Locality in Quantum Mechanics Portrayed as a Human Twins Metaphor" the author presents us with a description of the phenomenon of non-locality from a metaphor with twin brothers. With this, the author seeks to give us a more tangible view of this intriguing quantum phenomenon.

The author starts the manuscript with an elegant introduction where the history of quantum mechanics is detailed, starting with the works of Planck, Bohr (old quantum mechanics) and reaching the more formal description with the works of Schrödinger and Heisenberg. Then, the author discusses the peculiar aspects of quantum mechanics raised by Einstein, Podolsky and Rosen, and also by Schrodinger, which bring to light the non-local aspects of quantum theory. The introduction is then finalized by elucidating the fact that Euclidean space (3D) is not enough to describe the reality of matter, requiring the introduction of the spin of particles, as done by Dirac.

Then the author introduces a very peculiar story, of two identical twins, but one is left-handed and the other is right-handed. Soon after they are born it is totally impossible to distinguish one from the other by just observing them. However, when they begin to handle objects, it becomes possible to know which is left-handed and which is right-handed. So it is assumed that one of the twins is kept on Earth and the other is sent to Mars shortly after their birth. So, after a few years, if a geneticist on Mars discovers that the twin on Mars is right-handed, he will immediately know that the twin on Earth is left-handed, and vice versa. This situation is presented in the article to show that there is a correlation between the twins. But in this case, it is a classic correlation, as it is determined from the initial conditions. The author then makes it clear that there is nothing strange about it. However, the author then assumes that it is possible to determine a posteriori the characteristic of twins: for example, it assumes that somehow the characteristics of the twins are not pre-defined and that a geneticist is able to choose whether a given twin will be right-handed or left-handed a posteriori. With that, the author introduces what would be the equivalent of the quantum world in his example of twins: According to quantum mechanics, when a geneticist decides to choose that his twin will be right-handed (let's say on Mars), instantly the Earth's twin would have to become left-handed, without the need for a geneticist to do so and without the need for sending information and/or having an interaction between the Mars and the Earth twins. The author then points out how strange such a situation would be, which is what quantum theory predicts! The difference is that quantum theory is probabilistic, and this is discussed by the author. He even uses the fact that the measurement result is random exactly to show that it is not possible to use these quantum correlations to send messages faster than light.

In summary, we have a manuscript that discusses how strange the nonlocality of quantum mechanics is, being presented via an metaphory involving identical twins where one of them is left-handed and the other is right-handed. The paper is very didactic and illuminating, written in a language accessible to the general public. Therefore, I recommend its publication.

Author Response

I thank the referee for having read carefully the manuscript and that he did appreciate my narration about non-locality in quantum mechanics and the action at a distance, employing a metaphor involving human twins. I hope that my essay will also find the same interest to the readers of Entropy.

Reviewer 2 Report

In this article, the author presents a novel way to disseminate the phenomenon of quantum non-locality by using a metaphor involving human twins.

The article begins with an historic introduction to quantum mechanics, where the context of the article is given. I think it would be useful to say a few more words about the wave-function in this section, as it will be mentioned in the following ones. Furthermore, a few more citations are needed (for instance, the "historical paper by Werner Heisenberg" in note 2).

Then, the author presents the phenomenon of non-locality, which he begins to explain with a methaphor involving two twins which hand dexterity is entangled in a singlet state. Within this framework, he shows how the quantum features of entanglement are stronger than the classical ones and the fact that no superluminal transfer of information is admitted.

In this last part (from page 5 onwards), however, things start falling apart. The author explains a possible way to exploit entanglement to send messages, stating (rows 230-onwards) that it would be impossible to do so because it is forbidden by relativity (again, not cited). This explanation is not satisfactory and could easily confuse the reader. I strongly suggest to explain the reason why it is forbbidden in the same twins framework.

A similar critique can be done for the epilogue. There, the author states that many experiments confirmed the existance of entanglement, without providing even one citation. He then proceeds to briefly mention the effect of decoherence on entanglement in one sentence and then moves into the wave-particle duality argument, which was not explicitly introduced in the article. I find this ending quite confusing and I think it should be rewritten.

Author Response

I thank the referee for having read carefully the manuscript and to have found, correctly, some confusing passages along the narrative. I revised the whole text and amended it according to the recommended lines.

  1. I introduced more citations, passing from 18 in the first version to 43 in the present one, with several referencing the founders of quantum mechanics.
  2. I did rewrite the appointed paragraphs that seemed incomprehensible; I hope that now they became more intelligible after extending the explanation about why the action at distance disentangles instantaneously the parts, and why this issue remains out of the scope of special relativity.
  3. In the Epilogue, I introduced the references regarding recent experiments involving twin photons which, according to the authors of the papers, confirmed the “spooky action at a distance’’.
  4. About the decoherence issue, I inserted references to three revisions: one written by H.D. Zeh, who is the proponent of the theory, a comprehensive revision by W. H. Zurek, and a more recent one by M. Schlosshauer.
  5. About the particle-wave duality, my aim is to call attention to the fact that randomness and probabilities permeate all phenomena described by quantum mechanics, not only the random and instantaneous action at a distance. I rewrote the paragraph in order to keep a connection with the manuscript proposal.

Round 2

Reviewer 2 Report

In this revision, the author provides the argument about the no signaling and the citations that were missing in the early version.

There is still, in my opinion, some work to do on the conclusions: I don't really see the need of explaining the scientific method (lines 372-385), that space may be better used giving a brief explanation of the Bell experiments cited. Furthermore, when mentioning the Young experiment, I think it should be emphasized that it was done with a series of single electrons, briefly explaining how that shows the wave-particle duality.

Other small notes:
- line 202 is an empty in the middle of a paragraph;
- AAD is defined in note 9, but should be defined in the main text as well;
- "thence" in line 352.

Author Response

Thank you for your comments. Here below are the replies:

1. A brief explanation about Bohm and Bell contributions was inserted and
references were added.
2. A comment about Tonomura et al. experiment was added into the text after
referencing their paper.
3. The “scientific method” was removed from the body of the text, shortened
and inserted as a footnote.
4. The AAD acronym was inserted at first appearance into the manuscript.
5. Empty line 202, fixed.
